

# Analogous Hawking radiation in butterfly effect

**Takeshi Morita**[1,2*]

**1** Department of Physics, Shizuoka University,
836 Ohya, Suruga-ku, Shizuoka 422-8529, Japan
**2** Graduate School of Science and Technology, Shizuoka University,
836 Ohya, Suruga-ku, Shizuoka 422-8529, Japan

⋆ morita.takeshi@shizuoka.ac.jp

## Abstract

We propose that Hawking radiation-like phenomena may be observed in systems that show butterfly effects. Suppose that a classical dynamical system has a Lyapunov exponent $\lambda_L$, and is deterministic and non-thermal ($T = 0$). We argue that, if we quantize this system, the quantum fluctuations may imitate thermal fluctuations with temperature $T \sim \hbar\lambda_L/2\pi$ in a semi-classical regime, and it may cause analogous Hawking radiation. We also discuss that our proposal may provide an intuitive explanation of the existence of the bound of chaos proposed by Maldacena, Shenker and Stanford.

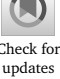

## 1 Introduction and Motivation

Prediction of thermal radiation from a quantum black hole is one of the outstanding achievement in theoretical physics [1, 2]. If the mass of the black hole is $M$, the temperature of the radiation is given by

$$T = \frac{\hbar c^3}{8\pi GM},\tag{1}$$

where we have set the Boltzmann constant $k_B = 1$. This temperature is proportional to the Plank constant $\hbar$, showing that the temperature arises from a purely quantum effect and it vanishes in classical mechanics ($\hbar = 0$).

Remarkably, such emergent thermal nature in quantum mechanics appears not only in black holes but also in various situations: Unruh effect in accelerated observers, acoustic Hawking radiation in supersonic fluids, moving mirrors and so on [3]. In these systems, quantum effects induce temperatures proportional to $\hbar$ similar to the Hawking temperature (1).

In this note, we review the proposal of Ref. [4] that an emergent quantum thermal nature may appear in dynamical systems that show butterfly effects. Let us consider a classical dynamical system that has a Lyapunov exponent $\lambda_L$[1]. We assume that this system at classical limit obeys deterministic dynamics and non-thermal. (We are in mind, for example, a driven pendulum motion.) We will argue that, once this system is quantized, the quantum fluctuations may imitate thermal fluctuations with temperature

$$T \sim \frac{\hbar}{2\pi}\lambda_L, \tag{2}$$

in a semi-classical regime. This temperature is proportional to $\hbar$ and it recovers $T = 0$ in the classical limit. Thus, it is analogous to the Hawking temperature (1), similar to the Hawking radiation-like phenomena mentioned above. Indeed, we will show that this emergent thermal property is related to acoustic Hawking radiation [5] in one-dimensional Fermi liquid [4,6,7].

Before going to discuss the details of our proposal, we briefly mention our motivation. Recently, Maldacena, Shenker and Stanford proposed the existence of the bound of chaos [8]. They considered quantum many body system at finite temperature and showed that the Lyapunov exponent of the system is generally bounded as

$$\lambda_L \leq \frac{2\pi T}{\hbar}, \tag{3}$$

where $T$ is temperature of the system. At the classical limit, this bound is trivial because the right hand side becomes infinity.

Here, we can rewrite this inequality as

$$T \geq \frac{\hbar}{2\pi}\lambda_L. \tag{4}$$

This relation tells us that the temperature of the chaotic system is bounded from below [9]. If we naively apply this inequality to classical chaotic systems that are non-thermal and deterministic, like a driven pendulum motion, it will lead to the following striking prediction. Suppose that the classical non-thermal chaotic system has a finite Lyapunov exponent $\lambda_L$. Then, the inequality (4) is satisfied trivially as $T = 0 \geq 0$. Here the temperature is zero because the system is non-thermal, and the right-hand side is also zero because $\hbar = 0$ in the classical limit. Thus, nothing is interesting so far. Now, we consider the quantum corrections to this relation, and ask what will happen in the semi-classical regime. Then, the right-hand side of the inequality (4) may become non-zero as

$$\frac{\hbar}{2\pi}\left(\lambda_L + O(\hbar)\right) = \frac{\hbar}{2\pi}\lambda_L + O(\hbar^2), \tag{5}$$

where we have assumed that the quantum corrections to the classical Lyapunov exponent $\lambda_L$ are $O(\hbar)$ [2]. Hence, if the bound (4) is correct, at least an $O(\hbar)$ temperature has to be induced in the system somehow quantum mechanically. It sounds like a Hawking radiation, and this prediction motivates us to study non-thermal chaotic systems in the semi-classical regime[3].

---

[1] In an $N$-body system in $d$ dimensional space that has a time reversal symmetry, we can define $2dN$ Lyapunov exponents $\{\pm\lambda_1, \cdots, \pm\lambda_{dN}\}$. Some of them may be complex. We assume that $\lambda_L$ is the maximal one and a positive real number.

[2] Although the definition of the Lyapunov exponent in quantum chaotic systems has not been established, the classical Lyapunov exponent would be well-defined semi-classically within the Ehrenfest's time. Indeed, our Hawking like phenomena occurs within this time [4, 10]. On the other hand, the out-of-time-order correlator (OTOC) [11] is actively investigated to compute the Lyapunov exponent in quantum mechanics [8, 12, 13]. For the application of OTOC to the inverse harmonic potentials, see Refs. [14, 15].

[3] We should emphasize that the application of the bound (4) to non-thermal systems is very rude, because the original work [8] was studied in finite temperature chaotic systems.

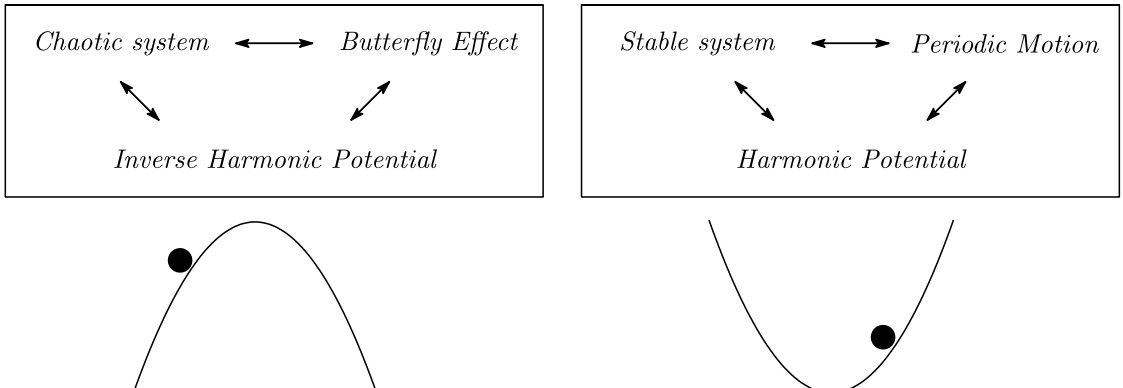

Figure 1: In chaotic systems, inverse harmonic potentials play fundamental roles to explain the butterfly effects. This is similar to the importance of harmonic potentials in stable systems.

Indeed, as we briefly mentioned around Eq. (2), an emergent thermal nature may appear in the systems that exhibit butterfly effects. Actually, butterfly effect is more essential than chaos in our study as we will see soon.

## 2 Butterfly Effect and Inverse Harmonic Potential

First, we briefly review butterfly effect in classical mechanics. This effect is a characteristic feature of chaotic systems. In a dynamical system, suppose that we observe a time evolution of a dynamical variable $q(t)$ for a given initial condition $q(0)$ at time $t = 0$. Then, by changing the initial condition slightly $q(0) + \delta q(0)$, we may observe the deviation of the dynamical variable $\delta q(t)$. If $\delta q(t)$ shows an exponential sensitivity to the initial condition,

$$\delta q(t) \sim \delta q(0) \exp(\lambda_L t), \qquad (\lambda_L > 0), \tag{6}$$

it is called the butterfly effect, since even very tiny fluctuation such as the flap of a butterfly may cause a huge deviation at late time. The exponent $\lambda_L$ in (6) is called the Lyapunov exponent. (See footnote 1 for more details.)

Naively, the exponential development (6) implies that the dynamical variable $\delta q(t)$ obeys the equation of motion,

$$\delta \ddot{q}(t) \sim \lambda_L^2 \delta q(t). \tag{7}$$

Hence, the motion of $\delta q(t)$ may be described by an effective potential

$$V_{\text{eff}}(\delta q(t)) \sim -\frac{1}{2} \lambda_L^2 \delta q(t)^2, \tag{8}$$

which is an inverse harmonic potential. Therefore, inverse harmonic potentials play important roles in butterfly effects. Actually, the tip of the inverse harmonic potential $\delta q(t) = 0$ is related to a hyperbolic fixed point in the context of dynamical system, and it is regarded as one of the essence of chaos [16]. See the sketch in FIG. 1. In the following arguments, we assume that some effective one-dimensional inverse harmonic potentials exist in butterfly effects, and consider quantum mechanics in these potentials.

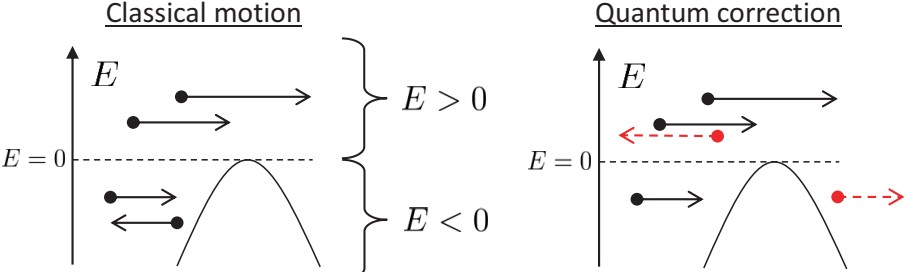

Figure 2: Particle motions in the inverse harmonic potential (9). The black arrows denote classical particle motions and the red dashed lines denote new particle trajectories in quantum mechanics that are forbidden in classical mechanics.

## 3  Emergent Thermodynamics in Inverse Harmonic Potential

We consider quantum mechanics in an one-dimensional inverse harmonic potential,

$$H = \frac{1}{2}p^2 - \frac{1}{2}\lambda_L^2 x^2, \tag{9}$$

where we have taken $x$ as the dynamical variable[4]. Obviously, this model exhibits the butterfly effect with the Lyapunov exponent $\pm\lambda_L$, $(\lambda_L > 0)$ as we have discussed in Sec. 2. We will first consider the classical motion. Then, we will see that the quantum corrections to the classical motion imitate thermal fluctuations and cause emergent thermal excitations[5].

Let us start from classical mechanics. Suppose free particles come from the left side of the potential ($x < 0$). See FIG. 2. The fate of these particles are very simple. If their energy $E$ are positive ($E > 0$), they go through the potential to the right side, and, if the energies are negative ($E < 0$), they are reflected by the potential and go back to the left side.

Next, we consider the quantum corrections to these motions. In the case of the negative energy particles $E < 0$, due to the quantum tunneling, the particles can penetrate the potential to the right side. Similarly, even in the case of the positive energy particles $E > 0$, the incoming particles may be reflected by the potential quantum mechanically. Therefore, new particle trajectories arise in quantum mechanics, which were forbidden in classical mechanics. These trajectories are sketched by the red arrows in FIG. 2.

We can exactly compute the probabilities for the occurrences of these new trajectories [4, 21, 22]. The tunneling probability $P_T(E)$ for the negative energy particle and the probability of the reflection of the positive energy particle are given by

$$P_T(E) = \frac{1}{\exp\left(-\frac{2\pi}{\hbar\lambda_L}E\right) + 1}, \quad (E < 0), \qquad P_R(E) = \frac{1}{\exp\left(\frac{2\pi}{\hbar\lambda_L}E\right) + 1}, \quad (E > 0), \tag{10}$$

respectively. Thus, we can combine these two formulas to

$$P(E) := \frac{1}{\exp\left(\beta_L|E|\right) + 1}, \qquad \beta_L := \frac{2\pi}{\hbar\lambda_L}. \tag{11}$$

---

[4]Classical particle motion in the inverse harmonic potential (9) is solvable. Hence, it is not chaotic, although it shows the butterfly effect. Therefore, butterfly effect is not a sufficient condition of chaos. As we show, the inverse harmonic potential is essential in our proposal for the emergent thermodynamics, and other chaotic properties are not relevant.

[5]Emergence of thermal natures in inverse harmonic potentials through quantum effects have been discussed in several contexts. See for example Ref. [6, 17–20].

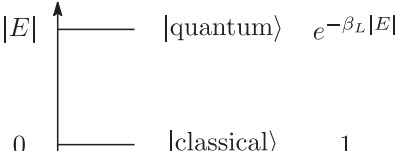

Figure 3: The sketch of the relation between the two trajectories (classical and quantum) and the probability ratio. The ratio of the probability of taking the classical trajectory to that of the quantum one is 1 to $\exp(-\beta_L |E|)$, where $E$ is the energy of the particle. It is identical to the thermal probability of a two level system, and taking the quantum trajectory corresponds to the excited state.

This result means that the ratio of the probability of taking the classical trajectory to that of the quantum one is 1 to $\exp(-\beta_L |E|)$. This ratio may be interpreted as the Boltzmann factor of the two level system at temperature,

$$T_L := \frac{1}{\beta_L} = \frac{\hbar}{2\pi} \lambda_L, \tag{12}$$

where the ground state (zero energy) and the excited state (energy $= |E|$ ) correspond to the classical trajectory and quantum one, respectively. See FIG. 3. Remarkably, the temperature $T_L$ saturates the bound of chaos (3) proposed by Maldacena, Shenker and Stanford [8]. Hence, our relation might be related to this bound.

So far, we have obtained the probability $\exp(-\beta_L |E|)$, which looks like a Boltzmann factor, but the connection to thermodynamics is unclear. We will see a clear interpretation by considering the energy transportation in the above processes.

In the case of $E < 0$, if the tunneling occurs, the negative energy particle is removed from the left side ($x < 0$), and thus the energy in this side increases by $-E(>0)$ comparing with the classical process. In the case of $E > 0$, if the quantum reflection occurs, the particle carrying the positive energy coming into the left side, and again the energy in the left region increases by $E$. Thus, in the both cases, the quantum corrections induce the energy $|E|$ in the left region with the probability $\exp(-\beta_L |E|)$. This is analogous to a thermal excitation! Taking the quantum trajectories that are forbidden in classical mechanics can be regarded as thermal excitations, which provide energy excesses $|E|$ in the left region over the classical processes.

On the other hand, if the particle takes the classical trajectory, the energy excess over the classical process is 0. Recall that the ratio of taking the classical trajectory to the quantum one is 1 to $\exp(-\beta_L |E|)$, and, since $1 = \exp(-\beta \times 0)$, it is consistent with the ground state of the two level system shown in FIG. 3.

In this way, the quantum corrections to the particle motions in the inverse harmonic potential can be regarded as the thermal excitations that cause the energy excess in the left side.

## 3.1 Relation to acoustic Hawking radiation

The energy transfer that we have seen is similar to Hawking radiation in black holes. Although any energy transfer from black hole to the outside does not occur in classical gravity, it does in quantum mechanics with the thermal probability[6].

The connection to Hawking radiation may be emphasized, if we consider many fermi particles in the inverse harmonic potential [4, 6, 7]. If we fill the inverse harmonic potential (9) with the right coming free fermi particles from below as shown in FIG. 4, the particles can be

---

[6]There are several studies that consider some relation between Hawking radiation and inverse harmonic potentials. See, for example, Ref. [19, 20, 23–26]

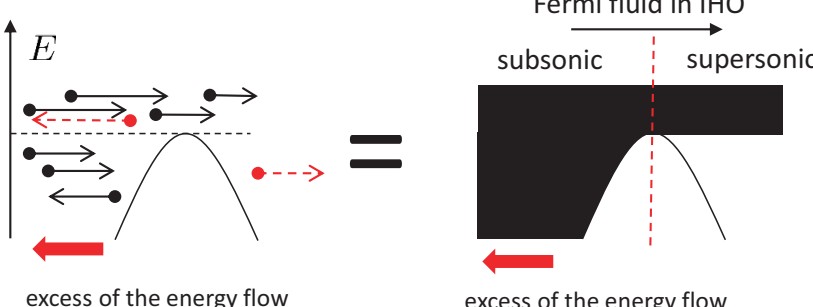

Figure 4: If we fill the inverse harmonic potential with many right moving fermions, the fermions can compose single fermi fluid. Then the energy transfer from the right side to left occurs in quantum mechanics comparing with the classical process. This energy transfer agrees with the acoustic Hawking radiation in the fermi fluid.

regarded as a single fermi fluid, which flows from the left[7]. Here, we have taken the fermi energy positive. Then, the thermal energy transfer from the right side to the left region occurs in quantum mechanics as we have discussed. Therefore, our quantum mechanics in the inverse harmonic potential predicts that the corresponding quantum energy flow occurs in the fluid mechanics too. We can show that this is precisely acoustic Hawking radiation [5]. It is not difficult to show that the fluid is subsonic in $x < 0$ and supersonic in $x > 0$, and acoustic Hawking radiation arises, once we quantize the phonon propagating on the fluid [4,6,7]. Thus, the particle motion in the inverse harmonic potential provides *the microscopic understanding* of the acoustic Hawking radiation in the fermi fluid.

## 3.2 Relation to the bound on chaos

Finally, we discuss a possible connection to the bound on chaos in quantum many-body systems at finite temperatures [8]. In the original work [8], the authors showed the existence of the bound on the Lyapunov exponent (3) by investigating the analytic properties of the OTOC. (See Ref. [27] for other approach.)

By applying our results on the analogous Hawking radiation in butterfly effect, we can explain why such a bound exists intuitively. Suppose that there are $N$ interacting classical particles in $d$-dimension at temperature $T$. Then, the system possesses $2dN$ Lyapunov exponents: $\pm\lambda_1, \cdots, \pm\lambda_{dN}$, where these exponents would depend on $T$. (We have assumed that the Hamiltonian is time reversal, and $\lambda_{dN}$ is the maximum one that is real and positive.) This system may have two time scales: the dissipation time $t_d$ and the scrambling time $t_*$ [8], and we may observe the exponential developments of the deviations of the observables $\delta X$ between these two time scales $t_d \leq t \ll t_*$. Thus, in this time scale, there is a mode $\delta X_{dN}$ which feels the effective potential $-\frac{1}{2}\lambda_{dN}^2 \delta X_{dN}^2$ causing the exponential development. Then, through the mechanism we have discussed, this mode would be disturbed quantum mechanically as if it has the temperature $T_{\text{eff}} \sim \frac{\hbar}{2\pi}\lambda_{dN}$. If $T \gg T_{\text{eff}}$, this effect may be irrelevant . However, if $T \ll T_{\text{eff}}$, the quantum fluctuations of $\delta X_{dN}$ may overcome the thermal fluctuations, and the thermal equilibrium state may be disturbed. Therefore, such a large Lyapunov exponent $\lambda_{dN}$ may be forbidden. It may intuitively explain the bound on chaos[8].

---

[7]We have assumed that the inverse harmonic potential is appropriately deformed so that the potential is bounded from below. For example, we replace it with a cos potential and focus around $x = 0$, See Ref. [10] for more details.

[8]Our result is consistent with Ref. [9], which argued that the bound of the Lyapunov exponent (3) may be saturated when the characteristic length scale of the chaotic system is the same order to the thermal de Broglie

# 4 Summary

In this article, we have argued the possibility of the emergent thermodynamics in butterfly effects. In our derivation, the assumption that the system has a mode that is effectively described by the one-dimensional motion in the inverse harmonic potential is crucial. Such a mode may be required to explain the butterfly effect (6), but it is not so obvious whether it is always true. Thus, it is important to investigate various chaotic systems and ask whether the emergent thermodynamics appear there.

One interesting application is the observation of the emergent thermodynamics in laboratories. Since butterfly effects are ubiquitous in our world, we might have a chance to observe them. Indeed, one proposal for an experimental observation has been argued in Ref. [28].

Finally, it is important to investigate the connection to black hole physics. Many people have argued that black hole must have a deep connection to chaos. Actually, the Lyapunov exponent of the black hole and the Hawking temperature saturates the bound of chaos [8]. Since we have argued that chaotic system may induce thermal radiation quantum mechanically, this induced one and Hawking radiation might be related. We hope to return this problem in the future.

## Acknowledgements

The author would like to thank Koji Hashimoto, Satoshi Iso and Gautam Mandal for valuable discussions and comments.

**Funding information** The work of T. M. is supported in part by Grant-in-Aid for Scientific Research C (No. 20K03946) and Grant-in-Aid for Young Scientists B (No. 15K17643) from JSPS.

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
