# Peer review of "Analogous Hawking Radiation in Butterfly Effect"

_SciPost Physics Proceedings, doi:SciPost Phys. Proc. 4, 007 (2021)_

## Round 1 · Referee Report · Anonymous (Referee 1) · 2020-12-13

Strengths

1-) the paper is well written and presents its main ideas clearly;

2-) the content of the paper is original and its main proposal is well-motivated via a few pedagogical examples;

3-) the paper helps to better understand the origin of the so-called bound on chaos, which is important in the context of black holes physics, many-body quantum chaos, condensed matter theory, etc.

Weaknesses

1-) In my opinion, the paper is already very good, but it could be even better if the author provides a more detailed comparison with previous results in the literature. In particular, I think the paper would greatly benefit from a more detailed comparison with the results of reference [9], by J. Kurchan.

Report

In the paper "Analogous Hawking Radiation in Butterfly Effect", Takeshi Morita proposes that in non-thermal systems that show sensitive dependence on initial conditions, quantum mechanics effects lead to the appearance of an effective temperature.

The author argues that the bound on chaos, $\lambda_L \leq \frac{2 \pi T}{\hbar}$, might be seen as a lower bound to the system's temperature: $T \geq \frac{\hbar \lambda_L}{2 \pi}$. In particular, the effective temperature introduced by quantum effects, $T_\text{eff} \sim \frac{\hbar \lambda_L}{2 \pi}$, defines a thermal scale below which quantum fluctuations overcome the thermal ones, and the thermal equilibrium may be disturbed. This provides a nice way to understand the chaos bound proposed by Maldacena, Shenker, and Stanford.

In my opinion, the paper is well written and presents original content. Therefore, I recommend the paper for publication after one minor clarification has been addressed.

Requested changes

1-)
I think the paper would greatly benefit if the author includes a more detailed comparison with previous results in the literature, in particular with reference [9], by J. Kurchan.
In [9], J. Kurchan explains the bound on chaos in a simple way. He defines a Lyapunov length scale, $\ell_0$, and argues that the bound on chaos appears when $\ell_0$ is comparable to the de Broglie length scale $\ell_\text{dB}$. Since, $\ell_0 \sim T^{1/2}$ and $\ell_\text{dB} \sim T^{-1/2}$, J. Kurchan's results seem to be consistent with the results obtained by Takeshi Morita, because for very low temperatures the de Broglie length scale becomes bigger than the Lyapunov length scale. I think the author could add a paragraph comparing his results with the ones in reference [9].

  • validity: good
  • significance: high
  • originality: high
  • clarity: high
  • formatting: excellent
  • grammar: good

Author:  Takeshi Morita  on 2020-12-16  [id 1083]

(in reply to Report 1 on 2020-12-13)
Category:
answer to question
correction

Dear Referee,

I would like to thank the referee for reading my manuscript and giving valuable comments. As the referee pointed out, my result is consistent with Ref.[9]. I added footnote 8 about this interesting relation to Ref.[9] at the end of section 3.2.

Sincerely,

Takeshi Morita

---

## Round 2 · Referee Report · Anonymous · 2020-12-26

Report

Since the author has addressed all my comments in the second version of his paper, I am happy to recommend the paper for publication.

---

## Round 2 · Author Response

Dear Referee,

I would like to thank the referee for reading my manuscript and giving valuable comments. As the referee pointed out, my result is consistent with Ref.[9]. I added footnote 8 about this interesting relation to Ref.[9] at the end of section 3.2.

Sincerely,

Takeshi Morita

---

## Round 2 · List of Changes

I added footnote 8 about the relation to Ref.[9] at the end of section 3.2.

You are currently on this page

Resubmission scipost_202010_00029v2 on 16 December 2020

---

## Editorial Decision

published